# Estimation of Sodium Availability and Food Sources from 2018 to 2019 and Its Trends during the 2004–2019 Period in Costa Rica

**DOI:** 10.3390/nu14153200

**Published:** 2022-08-04

**Authors:** Marlene Roselló-Araya, Karol Madriz-Morales, Jaritza Vega-Solano, Adriana Blanco-Metzler, Hilda Núñez-Rivas, Karla Benavides-Aguilar, Rafael Claro

**Affiliations:** 1Costa Rican Institute of Research and Education in Health and Nutrition (INCIENSA), Tres Ríos 4-2250, Costa Rica; 2Ministry of Health, San José 10101, Costa Rica; 3LINKS-Resolve to Save Lives Project, Fundación UCR-INCIENSA, San José 129-2100, Costa Rica; 4Nutrition Department (NUT), Federal University of Minas Gerais (UFMG), Belo Horizonte 30.140-100, MG, Brazil

**Keywords:** salt, sodium, population interventions, public health policies, food composition, Costa Rica, Latin America

## Abstract

Sodium availability and food sources in 2018–2019 were estimated and trends analyzed for 15 years (2004–2019) in Costa Rica. Food purchase records from the National Household Income and Expenditure Survey (ENIGH) 2018–2019 were converted to energy and sodium using food composition tables measuring “apparent consumption”. Foods were classified by sodium content. ENIGH is a probabilistic, stratified, two-stage and replicated national survey, carried out regularly by the national statistics institution. Results from the 2004–2005 and 2012–2013 ENIGHs came from previous analysis. Differences between periods were determined through descriptive and inferential statistics. The available sodium adjusted to 2000 kcal/person/day was 3.40, 3.86, and 3.84 g/person/day (g/p/d) for periods 2004–2005, 2013–2014, and 2018–2019, respectively. In this last period, this was 3.94 urban and 3.60 g/p/d rural (*p* < 0.05), with a non-linear increase with income. During 2004–2019 sodium from salt and salt-based condiments increased from 69.5 to 75.5%; the contribution of common salt increased, from 60.2 to 64.8% and condiments without added salt from 9.3 to 10.7%. From 2012–2013 to 2018–2019, processed and ultra-processed foods with added sodium intake increased from 14.2 to 16.9% and decreased in prepared meals (7.2 to 2.8%). Costa Rica has been successful in reducing salt/sodium available for consumption; after a 12% increase of salt consumption between 2004–2005 and 2012–2013, to a level almost twice as high as recommended, it has stabilized in the last period.

## 1. Introduction

There is a direct association between excessive sodium intake and the risk of developing high blood pressure (HBP) and other non-communicable diseases (NCDs) such as cardiovascular diseases (CVDs), chronic kidney disease, gastric cancer, osteoporosis [1,2,3] and dementia [4]. The Global Burden of Disease study showed that elevated sodium intake was the main CVD mortality and morbidity risk factor [5,6,7].

The World Health Organization (WHO) recommends an intake lower than 5 g (g) of salt (<2 g sodium) per adult per day and established the intake reduction goal of 30% by 2025 [5,6]. Most people consume up to more than twice the recommended daily intake [8].

In Costa Rica (CR), since the 1970s, CVDs represent the main cause of mortality, reaching 29% in 2018. On that same year prevalence of HBP in the population above 18 years old was 37.2% [9]. CR implemented the “National Strategy for the Comprehensive Approach to Chronic Non-Communicable Diseases and Obesity (ENTO) and its Action Plan” during 2013–2021. It included among its goals a 17% relative reduction in premature mortality from CVDs, cerebrovascular diseases and HBP, a 15% relative reduction of salt/sodium intake and a decrease in HBP’s relative increase [10]. The Development and Public Investment Plan 2019–2022 contemplates the decrease in the premature mortality rate due to NCDs, where the sodium consumption reduction may contribute to the improvement of the population’s health conditions [11].

Most ultra-processed foods (UP) have a high sodium content, and their supply, in terms of volume and variety, has spread widely throughout the whole world [12]. Sodium intake in CR’s adult population during the 2004–2005 period was 3.9 g/person/day (g/p/d) and in the 2012–2013 period was 4.6 g/p/d. This represents a 15% increase and twice the WHO’s maximum recommended intake. Main sodium sources were common salt (60%), processed foods, UP, and sodium-added condiments (27%) [13].

Interventions to reduce salt consumption are considered by WHO as “best-buys”, as they are among the most cost-effective measures that countries can implement to improve the population’s NCDs situation [5,14]. Reducing sodium consumption by implementing multiple interventions is possible, as demonstrated by Finland, England, and Brazil [15].

Sodium intake can be estimated through a wide range of methods. One of these, the Household Budget Survey (HBS) method has its strengths, but the most important limitation is that the survey was designed for economic reasons and not for nutritional purposes. Despite this, the Pan-American Health Organization (PAHO) considers, that the use of the HBS method represents a feasible option in countries with limited economic resources and data regarding food consumption [13,15,16].

This study analyzes the trend in sodium availability for consumption in three consecutive periods, using the same methodology so they are comparable. It is a pioneer in demonstrating that this availability for consumption stabilized, which is consistent with the actions carried out in the last decade by the National Plan for the Reduction of Salt/Sodium Consumption. The novelty is that the latest data available at the national level are analyzed, which makes surveillance sodium consumption possible (the first intervention of the SHAKE Package, which has been designed to assist Member States with the development, implementation, and monitoring of salt reduction strategies to enable them to achieve a reduction in population salt intake—https://www.who.int/publications/i/item/WHO-NMH-PND-16.4 guide) and changes in food consumption patterns. It facilitates the identification of food sources that should be focused on in intersectoral interventions. Its results are highly relevant for public health since it shows the urgent need for effective actions capable of maintaining stability in salt/sodium consumption and channeling resources for its reduction in the country.

These results are key to evaluate compliance with the goal established by the National Strategy for the Comprehensive Approach of NCDs and Obesity 2014–2021 [10] and the National Plan for Salt/Sodium Consumption Reduction in the Costa Rican Population 2011–2021 [17]. Likewise, it represents an essential input for the definition of the new sodium target for the NCD’s national strategy 2022–2030 and for the intervention prioritization for the action plan of this strategy to promote the modification of risky to health behaviors [18].

The present study estimated sodium availability and food sources during 2018–2019 and analyzed trends over 14 years (2004–2019) in the Costa Rican population.

## 2. Materials and Methods

The estimation of sodium available for consumption, food sources and time trends were performed using the indirect methodology first employed in Brazil based on the analysis of HBS data [16]. In CR, the HBS is known as the National Household Income and Expenditure Survey (ENIGH) and it is conducted regularly by the National Institute of Statistics and Census of CR (INEC) [19]. The ENIGH’s design is area probabilistic, stratified, two-stage and replicated for the domains: national, urban, and rural areas and planning regions. The selection unit is the housing, and the analysis unit the household and the people permanently residing in them. The total number of selected housing was 9828 and there is available information for 7046 households surveyed between February 2018 to March 2019. The survey involved 468 Primary Sampling Units (PSU), each of which was composed of a selection of 21 housing units, distributed across all the six planning regions.

PSU are the geographic areas built by the INEC Cartography Unit from census maps, searching for a size as homogeneous as possible according to the number of housing units: in urban areas, PSU have an average of 150 housing units, and in rural areas they have an average of 100 housing units.

Sample selection is representative by area, urban and rural, and at regional level, Central, Chorotega, Pacific Central, Brunca, Huetar Caribbean and Huetar North. A detailed description of the ENIGH sampling strategy is available on the website of INEC [19].

The ENIGH records every food and beverage purchase carried out by the household during seven consecutive days, later converted to a monthly figure. The expenditure record is “acquired”; this is a proxy measure known as “apparent consumption” and not “actual intake”. The income per household is net and includes that received per month during the last 12 months. Data were collected through interviews and self-completion of forms [19].

From the 2018–2019 ENIGH [19] expenditure for 878 foods was obtained; from which 686 foods whose consumption was greater than 0.01 g per day per person (g/d/p) were included, following the internationally standardized methodology [16] and the IDRC 1068888 research project guidelines [20], which are based on the consumed grams of the reported food. Fifty-one were added, for a total of 737 foods included in the 2018–2019 food composition table (FCT). Energy and nutrient values per 100 g of food, obtained through indirect and direct methods were included [21]. In order of importance, the sources used were the ValorNut System of the University of CR [22]; the National Nutrient Database for Standard Reference (USDA) [23]; the Centro America and Panama Nutrition Institute’s FCT [24]; the Colombian FCT [25]; nutritional information from commercialized foods’ labeling; calculations from recipes; and laboratory analysis results from research conducted by INCIENSA [20,26].

Foods were classified into five groups according to the sodium content, based on Monteiro et al. [27]: natural and processed foods without added sodium; common salt; condiments with added sodium; processed and UP foods with added sodium; and prepared dishes. The apparent sodium intake is showed in g/p/d for the total energy intake, and adjusted to 2000 kcal, by urbanization zone, planning region, and income quintile. The income quintiles correspond to the ordering of households according to their income in five groups of equal magnitude (20% each). The first quintile contains the 20% of households with the lowest gross income per capita, and the fifth the 20% of households with higher gross income per capita [28]. Adjusting the analysis to 2000 kcal allows comparisons on the same energetic base.

Trends in consumption of sodium food sources were estimated comparing the results of the analysis made in the 2004–2005 and 2012–2013 ENIGHs [13], with those of the present study (2018–2019). This information is comparable, except for exceptions due to updates in the food supply, as the same methodology was used [13]. A descriptive and inferential statistics analysis was performed to determine differences within periods. Student’s t-tests were used to compare each income quintile between the surveys, while a regression model (Generalized Linear Model) was used to investigate trends between the income quintiles within each survey. In each unit of analysis, the expansion factor was considered. This factor is obtained as the inverse of the probability of selecting each house at the time of selecting the sample. We tested the hypothesis (differences in the consumption of sodium between income quintiles and surveys) at the significance level of 5% by the Student’s t-test using the SPSS version 21.0 software [29].

## 3. Results

### 3.1. Estimation of Sodium Available for Consumption and Food Sources 2018–2019 in the Costa Rican Population

According to Table 1, the amount of sodium available for consumption, adjusted to 2000 kcal/person/day in 2018–2019 was 3.84 g/person/day, finding significant differences (*p* < 0.05) between urban (3.94 g/person/day) and rural areas (3.60 g/person/day). As per planning region, the Brunca, Central and Caribbean Huetar showed the highest sodium availability and the Chorotega the lowest, with significant differences between most regions (*p* < 0.05). Fifty percent of the regions surpass the national average sodium availability of 2000 kcal.

Table 2 shows that all income quintiles exceeded the maximum recommended sodium intake, the lowest being 3.43 and the highest 4.64 g/person/day adjusted to 2000 kcal/person/day (IV and III quintiles, respectively). There was no linear relationship found between income quintiles and sodium intake. Nonetheless, there was a statistically significant (*p* < 0.05) proportional linear relationship found between energy intake and income quintile.

As defined in Table 3, common salt was the main source (64.8%) followed by processed and UP foods with added sodium (16.9%) and condiments with added sodium (10.7%). Dietary sources that contributed the least were natural and processed foods without added sodium (4.7%) and prepared dishes (2.8%). Statistically significant differences (*p* < 0.05) between sodium food sources were found.

Among all dietary sources, significant differences were found within income quintiles, except for common salt between quintiles I and II and between IV and V (*p* > 0.05). In the highest income quintiles (IV and V), the main two sources were prepared dishes, processed and UP foods with added sodium. In the meantime, in the lowest income quintiles the two main sources were common salt and condiments with added sodium; while the least significant source was prepared dishes, purchased, and consumed at home. It is interesting that sodium in prepared dishes is 2.8%; however, in the lowest quintile it represents five times the contribution of the highest quintile.

### 3.2. Trends in Sodium Availability for Consumption and Food Sources in 2004–2005, 2012–2013 and 2018–2019

In the 2004–2005, 2012–2013 and 2018–2019 surveys, sodium availability for consumption adjusted to 2000kcal in Costa Rican households was 3.40, 3.86, and 3.84 g/p/d, respectively. A statistically significant (*p* < 0.0001) increasing trend was found among the first two periods, and in the last 2018–2019 survey, the availability of this nutrient showed a statistically significant (*p* < 0.05) 0.52% decrease when comparing it to 2012–013 (Figure 1).

In all three surveys, a significant difference (*p* < 0.05) was found in sodium availability in the rural area with respect to the urban area, with an increasing trend in the latter (Figure 2). In the rural area a decreasing trend was shown from 2012 to 2019 (*p* < 0.00).

Figure 3 compares sodium availability for consumption by income quintiles, with a minimum of 2.90 g/person/day in quintile V (2004–2005) and a maximum of 4.64 g/person/d in quintile III (2018–2019). There was no linear relationship with the income level. When comparing surveys, significant differences were found between all income quintiles (*p* < 0.05).

Sodium sources analyzed in all three surveys are shown in Figure 4. Common salt was the main source providing more than 60% and showing an increasing trend in 2018–2019, followed by the group of processed and UP foods with added sodium which also show a statistically significant (*p* < 0.05) increasing trend in the last two surveys. Condiments with added sodium are the third contributor, even though for the 2018–2019 survey there was a decrease compared to the 2012–2013 survey. The least contributing source was the prepared dishes group, while in the previous surveys (2004–2005 and 2012–2013) natural and processed foods without added sodium took this place.

In the 2004–2019 period, a nonlinear increase with income level was found in common salt contribution (60.2 to 64.8%). An increase in processed and UP foods with added sodium was observed (15.4 to 16.9%), as well as a decrease in fresh or processed and UP foods without added salt (5.4 to 4.7%) and prepared dishes (9.8 to 2.8%). In the 2012 to 2019 period, there was a reduction in condiments with added sodium (from 13.2 to 10.7%) and in prepared dishes (from 7.2 to 2.8%).

## 4. Discussion

Historically, salt has been considered a basic element to provide food with flavor, for food conservation and for food product formulation. Its consumption exceeds targets established by WHO and PAHO [5,6,7], which is particularly concerning given its association with HBP, CVD risk [2] and overall morbidity and mortality [2,6]. In CR this intake surpasses previously mentioned targets [5,6,7], not only nationally but also by geographic area and planning region. The average of sodium available for consumption in 2018–2019 was 3.84 g/person/day, matching what has been reported worldwide (65% consumes between 8–10 and 95% between 6–12 g of salt [30]), and what Carrillo-Larco et al. [31] have pointed out regarding sodium intake in Latin America and the Caribbean. The Caribbean Huetar region showed one of the highest sodium availabilities which could be related to its higher Afro-descendant population percentage in CR, globally associated with a higher HBP prevalence [32].

There was no linear relationship between income level (quintiles) and available sodium for consumption. However, for the first quintile this was 4.01 g/person/day adjusted to 2000 kcal, while for the fifth it was 3.50 g/person/day adjusted to 2000kcal. It is known that poor families’ low diet quality is greatly determined by the difference between monthly per capita consumption expenditure between high- and low-income families [13,33]. According to the last three ENIGHs, food expenditure in higher income households has been almost three times that for lower income households. Khandpur et al. [34] reported that exposure, availability, and access to higher sodium foods may explain the higher consumption in urban areas and higher income households.

In our study, only 2.8% of available sodium came from prepared dishes, this availability being five times higher in the highest income quintile than in the first quintile, most likely for cost or convenience reasons. Salleh et al. [35] reported Malaysian adults with higher socioeconomic status had a high sodium intake due to lack of time for cooking, which leads to a higher consumption of foods outside the home with high salt/sodium content. A French study [36] conducted with adults, revealed that low socioeconomic status populations, particularly women, dedicate more time to food preparation, and that, as well as education, social capital possession was associated with less frequency and time dedicated to meal preparation. In low socioeconomic status households’ diet is usually less diverse [37], and often there is a smaller variety of fruits and vegetables consumed when compared with high socioeconomical status households [38,39].

In 2018–2019, 75.5% of the sodium came from common salt and condiments with added sodium, corresponding with the findings of two evaluations conducted in CR [13]. Vadiveloa et al. [40] reported, in rural residents a higher use of salt and solid fats, and a lower use of vegetables and spices such as onion, oregano and sweet bell pepper in the preparation of beans. Likewise, when preparing rice, they used more salt and solid fat, and less oil, but more carrot and coriander. These results [40] are supported by other research that evidence that increasing the variety of herbs and spices to add flavor to foods can be particularly effective because it enhances palatability and reduces the use of less healthy ingredients such as fat and sodium [41,42,43,44]. Cooking foods with more salt has been associated with increased portion intake [40].

Almost one-sixth of the sodium comes from processed and UP foods with added sodium, up to 27% in the higher income households, as reported by Sarno et al. [15] in Brazil. If condiments with added salts are combined with these foods, then sodium from industrialized foods and ingredients increases to 27%. In high-income countries, most of the salt comes from processed foods, which has driven intervention approaches; whereas in low- and middle-income countries, such as CR, a community-based approach would be more realistic and effective since the main source is discretionary salt, added during cooking and/or at the table [2,45,46,47,48], followed by processed and UP foods with added sodium, whose consumption keeps rising [13,27].

The 2018–2019 ENIGH found a decreasing trend in apparent sodium intake at the national level, when comparing it to the 2012–2013 survey (from 3.86 to 3.84 g/person/day). Although the value continues to exceed the maximum recommended sodium intake per day per person [5,6,7], this is an important accomplishment for public health. This success is partly due to the launch in late 2012 of the salt/sodium intake reduction program [49], which has managed to interrupt and stabilize over a six-year period the increase in sodium intake. Moreover, the program had an impact on the salt/sodium available for consumption (0.52% reduction, *p* < 0.05).

Most countries have not managed to decrease the salt/sodium intake; modest reductions have been reported in middle- and high-income countries, for example, 25% in Finland, 14% in England and 12% in Brazil in 2008 [15]. The United Kingdom has successfully implemented a salt reduction program for over 10 years and achieved an intake reduction from 9.5 to 8.1 g/person/day (*p* < 0.05) in seven years, which means an average of 0.2 g of sodium per year. United Kingdom’s salt reduction program reduced its intake through a voluntary gradual reformulation [1]. CR is following United Kingdom’s example, as well as some of the strategies led by Japan and Portugal to achieve the population’s salt consumption reduction [50,51].

A higher sodium availability trend was found in the urban areas in the 2018–2019 survey (3.7%), while in the rural areas there is a decreasing trend (12.2%). These changes can be explained by the fact that urban areas present a greater exposure, availability, and access to foods high in salt, fat, and sugar in the food environment than that in rural areas [34,52].

Table salt continues to represent the main sodium source in the diet of CR households, providing more than 60% of the total with an increasing trend in the studied period; followed by processed and UP foods with added sodium, which also show a statistically significant (*p* < 0.05) increasing trend when compared to surveys conducted in 2012–2013, 2018–2019. Similar results have been reported in Asian countries [53,54]. The International cross-sectional study of macronutrients and micronutrients and blood pressure (INTERMAP) is a basic epidemiological research study of 4680 men and women aged 40 to 59 years from 17 diverse population samples. This study reported that in the United Kingdom (5%), China (7.6%), Japan (9.5%) and USA (29%) of sodium was attributed to salt added while cooking at home [55]. These results agree with the trends reported in Brazil [15], where table salt and salt-based condiments are the main sodium source (74.4%). In Mexico, processed and UP foods have become one of the main dietary sources, providing between 39% and 50% of the sodium intake, followed by table salt (36%) [56]. Strong associations between UP food consumption and HBP have been shown [12,57,58,59,60], which is why promoting unified policies that impact the reduction of the consumption of these foods and encourage a healthier diet is urgent [61].

On the other hand, gradual increases in the intake of high-salt processed foods suggests that a dietary and lifestyle behavior change is necessary [62,63], and insists on the need for formative research to design community-based approaches. The WHO STEPwise approach to NCD risk factor surveillance (STEPS) is a simple, standardized method for collecting, analyzing, and disseminating data on key NCD risk factors in countries. In Nepal, STEPS surveys (2013 and 2020) reported that high sodium processed food intake increased from 11.5 to 19.5% [64].

Condiments with added sodium represented the third sodium contributor to Costa Rican population’s diet between 2014 and 2019. Yet, in the 2018–2019 survey, its contribution was of 10.7% and it decreased compared to 2012–2013 [13], perhaps because the use of salt increased by almost 5%, displacing it to third place. The prepared dishes group contributed the least to dietary sodium in this survey (2018–2019), whilst in previous surveys natural and processed foods without added sodium took this place. The latter group supplied less than 5% in the 2018–2019 survey, proving that its purchase does not represent a habit in Costa Rican households, an undesirable trend also reported in the USA [65].

Despite the health benefits of consuming natural foods, their intake remains low, and one of the reasons is taste perception which, in turn, influences choice and intake [66]. The final taste of a food is the result of a combination of flavor, odor, and chemical reaction [67]. According to Douglas A et al. [68], the use of a blend of herbs and spices to boost the flavor and palatability of foods has the potential of helping consumers reduce salt intake and improve natural food acceptance.

Among the limitations of the present study appears the fact that ENIGH or HBS is conducted for economic analysis purposes to determine monthly households’ expenditures in the acquisition of goods and services; therefore, it does not pursue nutritional objectives. This method overestimates sodium consumption because it assumes that all purchased foods and beverages are for human consumption at home. On the other hand, it does not consider what is consumed outside the home, underestimating real intake; for that reason, it is not possible to obtain real total sodium intake, and thus apparent intake is estimated. Besides, as reported by Blanco-Metzler et al. [13], FCT do not accurately record data on food sodium content, given that it depends on the data source. In CR, most of the data on sodium content comes from the USDA’s FCT, which is in turn ValorNut’s [22] main data source. There is a lack of updated composition data for locally consumed foods; data available has been provided by INCIENSA [20,26]. Another limitation are the changes in foods recorded in the surveys, and the re-classification of some of the different groups examined, so these aspects must be considered when comparing the results between periods.

It is crucial to estimate sodium intake in Costa Rican population with more accurate methods, like the 24-hour urinary sodium excretion (gold standard), even though the HBS method allows a rough estimation and enables the study of trends in sodium availability in households [15,16]. This ecological method has the advantage of being inexpensive, accessible and fast, and data is available and generated systematically. This also allows the monitoring of apparent intake throughout time and the impact of interventions implemented, as well as dietary sources; the latter aspect is not possible using 24-hour urinary sodium excretion.

CR has faced ongoing challenges in the development and implementation of actions to reduce excessive salt intake. These include lack of salt/sodium intake data, limited commitment of food industries and opposition from salt companies, as well as lack of effective political feasibility for the strengthening and sustainability of actions. These challenges should not hinder the continuity of interventions to impact prevention of HBP and CVDs.

To strengthen the stabilization evidenced in the present study, and make it sustainable, it is suggested that salt reduction strategies should be adapted to the local context, keeping in mind the differences in dietary and lifestyle-related behaviors [62,63] and sodium’s main sources [62], such as salt in CR’s case. It is necessary to change food preparation patterns and to promote the development and consumption of low-sodium foods [14,69,70]. Likewise, elements of local participation are critical to involve community members in the formative stage and explore social and cultural drivers of excessive salt/sodium intake. With this type of co-design, the community’s support and enthusiasm is drawn and sustainable reductions in salt consumption are promoted [71,72].

## 5. Conclusions

From 2013–2014 to 2018–2019, sodium availability for consumption remained stable, after increasing 12% from 3.40 to 3.86 g/person/day/ from 2004–2005 to 2013–2014. The results obtained in the last survey indicate that the efforts made by CR after 2011 to reduce the excessive consumption of salt/sodium in the population have been successful, since its consumption has stabilized. Nonetheless, the sodium food sources pattern remains with increasing trends towards processed and UP foods intake; a crucial aspect to consider in public health policies, in order to encourage the updating of the sodium reduction targets in prepackaged foods, mandatory nutritional labeling, frontal warning labeling, and regulation of the marketing of prepackaged foods with excess sodium.

A culturally adapted integrated policy that includes a community-based awareness and training program is needed to motivate people to use less salt. Additionally, to achieve the 30% mean salt intake reduction goal by 2025, political will and collaborative work between the health sector, food industry, gastronomic sector and retail points must be strengthened.

## Figures and Tables

**Figure 1 nutrients-14-03200-f001:**
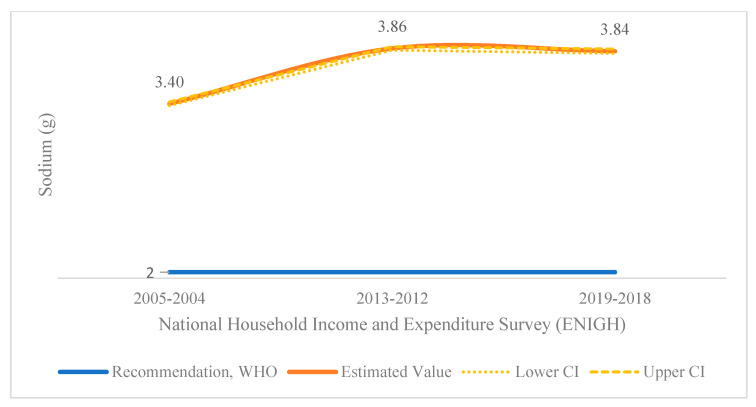
Sodium availability for household consumption in Costa Rica during 2004–2005, 2012–2013 and 2018–2019 (g/person/day adjusted to 2000 kcal/day). Lower CI = lower limit of the confidence interval. Upper CI = upper limit of the confidence interval.

**Figure 2 nutrients-14-03200-f002:**
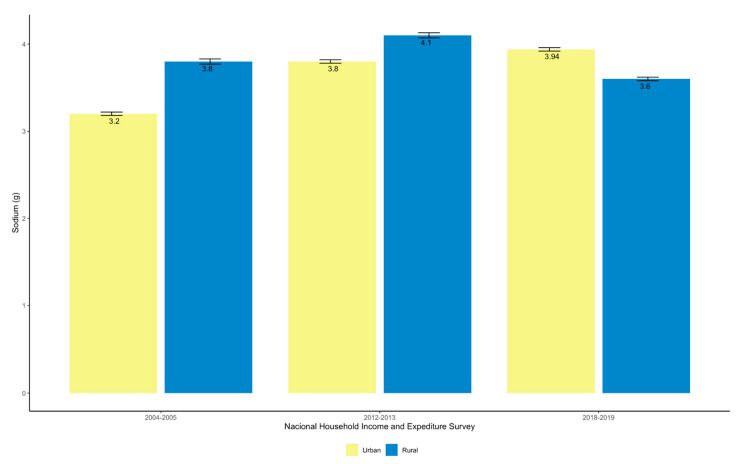
Sodium availability for household consumption according to zone of residence in Costa Rica during 2004–2005, 2012–2013 and 2018–2019 (g/day/person adjusted to 2000 kcal/day).

**Figure 3 nutrients-14-03200-f003:**
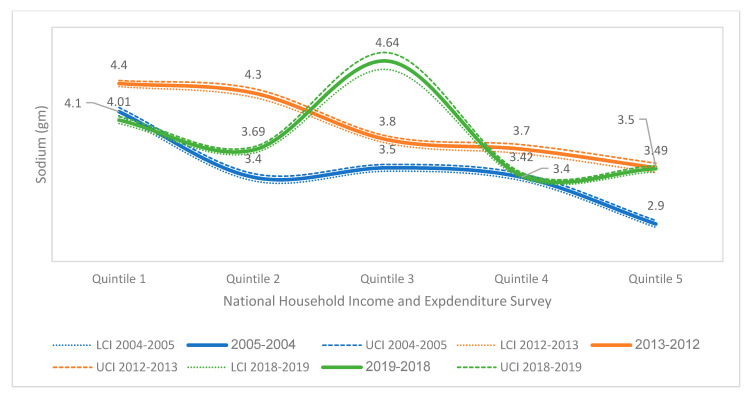
Sodium availability for household consumption by income quintile in Costa Rica during 2004–2005, 2012–2013 and 2018–2019 (g/person/day adjusted to 2000 kcal/day). LCI = lower limit of the confidence interval. UCI = upper limit of the confidence interval.

**Figure 4 nutrients-14-03200-f004:**
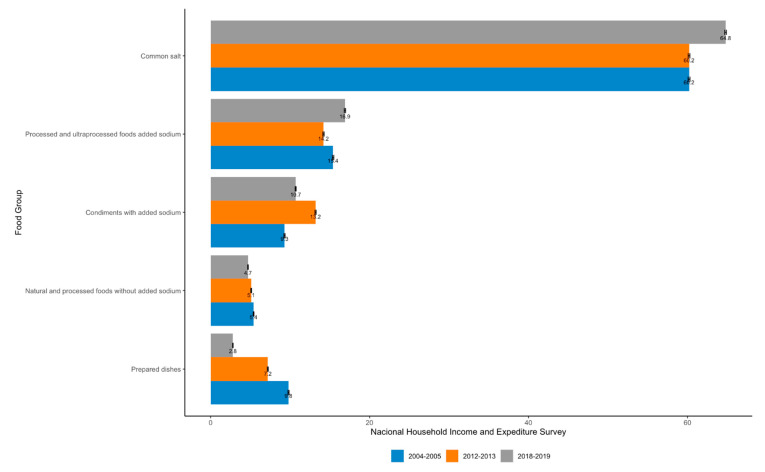
Dietary sources of sodium in Costa Rican households (%) by food group and National Household Income and Expenditure Survey 2004–2005, 2012–2013 and 2018–2019.

**Table 1 nutrients-14-03200-t001:** Energy and sodium availability for household consumption at the national level, by zones and planning regions in Costa Rica, 2018–2019.

	Total Energy (kcal/Person/Day)	Sodium in Total Energy (g/Person/Day)	Sodium Adjusted to 2000 kcal (g/Person/Day)
National	2302	4.42	3.84
**Zone**			
Urban	2252 a	4.44 a	3.94 a
Rural	2422 b	4.37 b	3.60 b
**Region**			
Central	2239 c	4.39 c	3.92 c
Chorotega	2326 d	3.79 d	3.26 d
Central Pacific	2397 cd	4.39 cd	3.66 cd
Brunca	2384 e	4.68 e	3.93 e
Caribbean Huetar	2472 f	4.81 f	3.90 f
North Huetar	2418 e	4.50 e	3.72 e

a,b,c,d,e,f: different letters within the same column indicate significant differences (*p* < 0.05) between areas and regions. For zone by t-Student; region by Anova and Tukey’s HSD.

**Table 2 nutrients-14-03200-t002:** Energy and sodium availability for household consumption based on food purchases by increasing quintiles of income distribution in Costa Rica, 2018–2019.

Quintile of per Capita Income	Population	Total Estimated Energy (kcal/Person/Day) *	Sodium in Total Estimated Energy (g/Person/Day) *	Sodium Adjusted to 2000 kcal (g/Person/Day) *
I	307,821	1905 a	3.82 a	4.01 a
II	307,709	2187 b	4.04 b	3.70 b
III	307,978	2327 c	5.39 c	4.64 c
IV	307,497	2451 d	4.20 d	3.43 d
V	307,699	2641 e	4.63 e	3.50 e

* a,b,c,d,e: different letters within the same column indicate significant differences (*p* < 0.05) between each quintile. For quintile by Anova and Tukey’s HSD.

**Table 3 nutrients-14-03200-t003:** Dietary sources of sodium in households according to income quintile, Costa Rica, 2018–2019.

Food Group	Costa Rica			Quintiles of per Capita Income Distribution (%) **				
	Population	g/Person/Day *	% *	I	II	III	IV	V
Natural and processed foods without added sodium	1,538,704	0.18 a	4.7 a	16.0 a	18.3 b	19.3 c	21.8 d	24.6 e
Common salt	1,538,704	2.49 b	64.8 b	23.6 a	20.1 a	25.8 b	16.0 c	14.4 c
Condiments with added sodium	1,538,704	0.41 c	10.7 c	17.2 a	18.7 b	25.4 c	18.2 d	20.5 e
Processed and ultraprocessed foods with added sodium	1,538,704	0.65 d	16.9 d	15.5 a	17.0 b	18.6 c	22.4 d	26.5 e
Prepared dishes	1,538,704	0.11 e	2.8 e	7.5 a	12.8 b	16.7 c	25.1 d	37.9 e

* a,b,c,d,e: different letters within the same column indicate significant differences between food groups (*p* < 0.05). ** a,b,c,d,e: different letters within the same row in the case of quintiles, indicate significant differences (*p* < 0.05). For quintile by Anova and Tukey’s HSD.

## Data Availability

The ENIGH 2018 data are publicly accessible, registered at: http://sistemas.inec.cr/pad5/index.php/catalog/22.

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
