# Peer review of "Estimation of Sodium Availability and Food Sources from 2018 to 2019 and Its Trends during the 2004–2019 Period in Costa Rica"

_nutrients, 2022, doi:10.3390/nu14153200_

Round 1

Reviewer 1 Report

The paper “Estimation of sodium availability and food sources from 2018 to 2019 and its trends during the 2004-2019 period in Costa Rica” contributes to the growth of literature for research on eating behaviour,  especially the consumption of sodium.

Before the manuscript acceptation for publication in “Nutrients” the following items should be revised:

Introduction

What are the weaknesses of this method?

It is no clear description of the aim of the research

Methods

What was the method of selecting the group?  (The current population of Costa Rica is 5,187,863)

Results

Table 1-3      a,b,c,d,e: different letters within the same column indicate significant differences between food groups (p<0.05). - what test was calculated

Figure 1–4  The authors should mark error bars.

Figure 3  There is no explanation of the terms (I -V) in the title and the chart.

Conclusion

What are the limitations of this method?

Author Response

Comment: The paper “Estimation of sodium availability and food sources from 2018 to 2019 and its trends during the 2004-2019 period in Costa Rica” contributes to the growth of literature for research on eating behaviour, especially the consumption of sodium.

Before the manuscript acceptation for publication in “Nutrients” the following items should be revised:

  • Introduction

What are the weaknesses of this method?

It is no clear description of the aim of the research

Response: Thank you very much for these comments. We agree that these are very important considerations, and we have added them. These are in red font in the manuscript.

  • Methods

What was the method of selecting the group?  (The current population of Costa Rica is 5,187,863).

Response: These aspects are expanded in the section on materials and methods. They are in red font in the manuscript.

  • Results

Table 1-3      a,b,c,d,e: different letters within the same column indicate significant differences between food groups (p<0.05). - what test was calculated

Figure 1–4  The authors should mark error bars.

Figure 3  There is no explanation of the terms (I -V) in the title and the chart.

Response: Thank you very much for these observations. We agree that these are very important considerations, and we have added them in tables 1-3 and figures 1-4. They are in red font in the manuscript.

  • Conclusion

What are the limitations of this method?

Response: Thank you very much for this comment. We have added it. This is in red font in the manuscript.

Reviewer 2 Report

The present study aims at providing insights on the sodium availability and food sources from Costa Rica between the years 2018 to 2019 and its trends during 2004-2019 period. A total number of households over 7000 were surveyed in order to obtain the results of the study. The statistical analysis of the study is done by descriptive and inferential analysis. The topic of the present paper is relevant and may be of interest for the readers of the journal.

I believe that the Introduction concisely sets up the following sections of the manuscript, including the objectives of the study, which are clearly stated and identified.

On the other hand, even though the authors provided the interpretation of the obtained results in a well written manner, I am unconvinced that the employed statistical analysis is the suitable approach and the number of households are relevant for the scope of the study in order to draw a valid conclusion.

Author Response

The present study aims at providing insights on the sodium availability and food sources from Costa Rica between the years 2018 to 2019 and its trends during 2004-2019 period. A total number of households over 7000 were surveyed in order to obtain the results of the study. The statistical analysis of the study is done by descriptive and inferential analysis. The topic of the present paper is relevant and may be of interest for the readers of the journal.

I believe that the Introduction concisely sets up the following sections of the manuscript, including the objectives of the study, which are clearly stated and identified.

On the other hand, even though the authors provided the interpretation of the obtained results in a well written manner, I am unconvinced that the employed statistical analysis is the suitable approach and the number of households are relevant for the scope of the study in order to draw a valid conclusion.

Response: These aspects are expanded in the section on materials and methods. They are in red font in the manuscript.

Reviewer 3 Report

Line 26: "%" must be added after 14.2 - 16.9

Line 28-29: too many words in Keywords

"Quintiles" which is first shown in Table 2, should be better briefly explained  in Material & Methods.

INTERMAP study (line 274) and STEPS survey (line 286) should be better briefly explained. 

Author Response

Comment

Line 26: "%" must be added after 14.2 - 16.9

Line 28-29: too many words in Keywords

"Quintiles" which is first shown in Table 2, should be better briefly explained in Material & Methods.

INTERMAP study (line 274) and STEPS survey (line 286) should be better briefly explained. 

Response: All observations were made. Some keywords were removed. Materials and methods section briefly describe what corresponds to income quintiles. The discussion expands on the INTERMAP study and the STEPS survey. They are in red font in the manuscript.

Reviewer 4 Report

This is a relevant report, both for policy makers in Costa Rica and for a broad readership interested in salt intake levels. The current abstract ends with stating:

 Costa Rica has been successful in the  reduction of excessive intake of salt/sodium, achieving a downward trend. I strongly advise to conclude differently: From 2013-2014 to 2018-2019 the salt intake remained stable, following an increase from 3.40 to 3.86 g/p/d/ from 2004-2005 to 2013-2014. Reasons to delete this too positive statement are,

-        the very minor decrease (0,58% , 0.02g sodium) /d. Not really significant and relevant, comparing e.g. to the reduction realized in the UK of 1.4g salt/day ~0,6 g sodium/day.

-        the exclusion of outdoor eating –  an increasing trend around the globe, and usually with higher salt levels than used at home (any data available for Costa Rica?)

-        the limited accuracy of the applied methods

The text of the paper should mention these considerations.

I presume that food policy makers will highly appreciate when some of the most consumed food products and condiments accounting for a significant part to the salt intake can be specified. For instance in the United Kingdom, bread was identified as a key contributor to salt intake; therefore (in fact almost mandatory) reductions of the target salt levels in bread were  issued by the Food Standard Agency.

The abstract should be rewritten and should include at least

-        the 3 levels of average sodium intake in 2004-5, 2013-14 2018-19

-        the percentages of salt (or sodium)supplied by – salt as such, condiments and food products

-        the limitation of the methods used (no outdoor eating included)

-        the adapted conclusion (stabilization – and not suggesting a decrease.)

-         

Table 2, Table 3.   Explain in the text below the table all symbols used in the table - - also n and I up to V - and not only a,b,c,d,e,

Figure 3 Can an explanation be given for the exceptional increase in salt intake in 2013-14 , only for Group iii.?

All figures: can the level of uncertainty be indicated with vertical (Fig 1.2.3) and horizontal (Fig 4) E.g. at the 0,01% level?

Author Response

Comment

This is a relevant report, both for policy makers in Costa Rica and for a broad readership interested in salt intake levels. The current abstract ends with stating:

  • Costa Rica has been successful in the reduction of excessive intake of salt/sodium, achieving a downward trend.I strongly advise to conclude differently: From 2013-2014 to 2018-2019 the salt intake remained stable, following an increase from 3.40 to 3.86 g/p/d/ from 2004-2005 to 2013-2014. Reasons to delete this too positive statement are,

-        the very minor decrease (0,58%, 0.02g sodium) /d. Not really significant and relevant, comparing e.g. to the reduction realized in the UK of 1.4g salt/day ~0,6 g sodium/day.

-        the exclusion of outdoor eating –  an increasing trend around the globe, and usually with higher salt levels than used at home (any data available for Costa Rica?)

-        the limited accuracy of the applied methods

The text of the paper should mention these considerations.

I presume that food policy makers will highly appreciate when some of the most consumed food products and condiments accounting for a significant part to the salt intake can be specified. For instance in the United Kingdom, bread was identified as a key contributor to salt intake; therefore (in fact almost mandatory) reductions of the target salt levels in bread were  issued by the Food Standard Agency.

The abstract should be rewritten and should include at least

-        the 3 levels of average sodium intake in 2004-5, 2013-14 2018-19

-        the percentages of salt (or sodium) supplied by – salt as such, condiments, and food products

-        the limitation of the methods used (no outdoor eating included)

-        the adapted conclusion (stabilization – and not suggesting a decrease.)

Response:  We appreciate this suggestion, which was considered to rewrite the abstract. They are in red font in the manuscript.

  • Table 2, Table 3.   Explain in the text below the table all symbols used in the table - - also n and I up to V - and not only a,b,c,d,e,
  • Figure 3 Can an explanation be given for the exceptional increase in salt intake in 2013-14, only for Group iii.?
  • All figures: can the level of uncertainty be indicated with vertical (Fig 1.2.3) and horizontal (Fig 4) E.g. at the 0,01% level?

Response: Thank you very much for these observations. We agree that these are very important considerations, and we have added them in tables 1-3 and figures 1-4. They are in red font in the manuscript.

Reviewer 5 Report

This article is well written, however, some comments to be considered during revision are:

1. In Abstract, please add the brief Method during this study.

2. The gap analysis is highlighted in Introduction, however, the novelty of this study is not clearly stated.

3. The specific objective of this study is suggested to be added in the last part of Introduction.

4. The statistical test for significant analysis in table 2 and Table 3 should be included in Table caption.

5. The discussion must be compared with other nations (if available).

Author Response

Comment:

This article is well written, however, some comments to be considered during revision are:

  1. In Abstract, please add the brief Method during this study.

Response:  We appreciate this suggestion, which was considered to rewrite the abstract. They are in red font in the manuscript.

  1. The gap analysis is highlighted in Introduction, however, the novelty of this study is not clearly stated.

Response: We appreciate this suggestion, which was considered. It is in red font in the manuscript.

  1. The specific objective of this study is suggested to be added in the last part of Introduction.

Response: Thank you very much for this observation. The objective moved at end of introduction. It is in red font in the manuscript.

  1. The statistical test for significant analysis in table 2 and Table 3 should be included in Table caption.

Response: Thank you very much for these observations. We agree that these are very important considerations, and we have added them in tables 2 and 3. They are in red font in the manuscript.

  1. The discussion must be compared with other nations (if available).

Response: We appreciate this suggestion; however, the authors consider that the comparison with other countries is broad and up-to-date.

Round 2

Reviewer 2 Report

The authors of the article “Estimation of sodium availability and food sources from 2 2018 to 2019 and its trends during the 2004-2019 period in Costa 3 Rica” addressed all the suggestions made and managed improve the research article. On the other hand, Figure 2 and Figure 4 must be uploaded in a higher resolution because in the present form the text is not clear.

Author Response

Response: Thank you very much for this observation. When including the graphics in the Word manuscript, the resolution does not seem clear. These graphs were elaborated in the R program and in .jpeg format they are better appreciated. We tried to attach the figures in .jpeg format but the system only allows Word and PDF files.
The four figures of the manuscript were sent by email in a .zip file to Ligia Cimpean ([email protected]) so that she can send it to you.

Thanks,

Reviewer 4 Report

Thank you for the major improvements made. However, the final statement  made in the abstract and in the conclusions is still incorrect and misleading. " " Costa Rica has been successful in reducing salt/sodium consumption, achieving a downward trend, which remained stable in the last period."  

the reduction of 0,02g Na/day is nutritionally negligible - as is the downward trend. It even may be incorrect, since outdoor eating, a growing trend worldwide, often with more processed foods and higher salt levels, was not taken into account. Please use a statement without 'downward', such as: "After a 12% increase of salt consumption between 2004-2005 and 2012-2013 to a level almost twice as high as recommended has stabilized since then" . IF this can be substantiated in some way one may add that stabilization may have been realized due to the actions resulting from Costa Rica's  2011-2021 National Plan for Salt/Sodium Consumption Reduction. 

In the text - pls clarify the meaning of CI (Fig. 1) and LCI and UCI (Fig. 3). 

Finally - it is regrettable that next to the clear indications of sources of salt:  salt as such and in condiments, the vague term of (ultra) processed foods is  used, instead of naming key processed products with high salt levels by name - such as for example ' pizza' or ' bread' 

the  downward trend in salt consumption is still mentioned:  I urge you to change this, taking into acccount my earlier remarks - even when the very minor decrease of 0,02g Na/ d  consumption may be statistically significant - it is irrelevant for nutrition and health. Also Such a minor reduction may also be 'compensated' by the often higher salt content of outdoor eating (not included in the study) 

Author Response

Reviewer 4

Comment: Thank you for the major improvements made. However, the final statement made in the abstract and in the conclusions is still incorrect and misleading. " " Costa Rica has been successful in reducing salt/sodium consumption, achieving a downward trend, which remained stable in the last period." 

- the reduction of 0,02g Na/day is nutritionally negligible - as is the downward trend. It even may be incorrect, since outdoor eating, a growing trend worldwide, often with more processed foods and higher salt levels, was not taken into account. Please use a statement without 'downward', such as: "After a 12% increase of salt consumption between 2004-2005 and 2012-2013 to a level almost twice as high as recommended has stabilized since then". IF this can be substantiated in some way one may add that stabilization may have been realized due to the actions resulting from Costa Rica's  2011-2021 National Plan for Salt/Sodium Consumption Reduction.

In the text - pls clarify the meaning of CI (Fig. 1) and LCI and UCI (Fig. 3).

Finally - it is regrettable that next to the clear indications of sources of salt:  salt as such and in condiments, the vague term of (ultra) processed foods is used, instead of naming key processed products with high salt levels by name - such as for example ' pizza' or ' bread'  

the  downward trend in salt consumption is still mentioned:  I urge you to change this, taking into acccount my earlier remarks - even when the very minor decrease of 0,02g Na/ d  consumption may be statistically significant - it is irrelevant for nutrition and health. Also Such a minor reduction may also be 'compensated' by the often higher salt content of outdoor eating (not included in the study)

Response: Thank you very much for these comments. We agree that these are very important considerations, and we have added them in abstract, conclusion and figures 1 and 3.

Regarding your comment “…the vague term of (ultra) processed foods is used, instead of naming key processed products with high salt levels by name - such as for example ' pizza' or ' bread’”  in accordance with the objectives of this study, the results were analyzed by food groups (Monteiro classification) and not by specific foods. This analysis by food source will be done for a later manuscript.
